# Text Representation Distillation via Information Bottleneck Principle

**Yanzhao Zhang**[1]**, Dingkun Long**[1]**, Zehan Li, Pengjun Xie**[1]
[1]Alibaba Group
{zhangyanzhao.zyz,dingkun.ldk,pengjun.xpj}@alibaba-inc.com

## Abstract

Pre-trained language models (PLMs) have recently shown great success in text representation field. However, the high computational cost and high-dimensional representation of PLMs pose significant challenges for practical applications. To make models more accessible, an effective method is to distill large models into smaller representation models. In order to relieve the issue of performance degradation after distillation, we propose a novel Knowledge Distillation method called **IBKD**. This approach is motivated by the Information Bottleneck principle and aims to maximize the mutual information between the final representation of the teacher and student model, while simultaneously reducing the mutual information between the student model's representation and the input data. This enables the student model to preserve important learned information while avoiding unnecessary information, thus reducing the risk of over-fitting. Empirical studies on two main downstream applications of text representation (Semantic Textual Similarity and Dense Retrieval tasks) demonstrate the effectiveness of our proposed approach[1].

## 1 Introduction

Text representation is a crucial task in natural language processing (NLP) field that aims to map a sentence into a single continuous vector. These representations can be applied to various downstream tasks, such as semantic textual similarity (Agirre et al., 2016; Reimers and Gurevych, 2019), information retrieval (Karpukhin et al., 2020; Long et al., 2022a), text classification (Garg et al., 2020), etc. Pre-trained language models (PLMs) have recently become the dominant approach for text representation. However, text representation models developed on large PLMs typically require enormous computational resources and storage that prevent

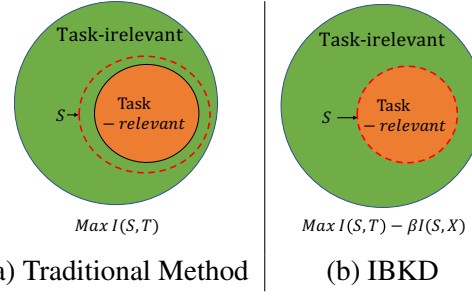

Figure 1: Information diagram of teacher embeddings (original circle), student embeddings (dashed red circle) and original input X (green circle). The left side represents the final optimization object of the traditional method, while the right side depicts the object of the IBKD method.

widespread deployment. Further, direct training on small-scale PLMs will lead to a significant decrease in model performance (Zhao et al., 2022).

Knowledge distillation (KD) is a commonly used approach to reduce the performance gap between large and small models (Hinton et al., 2015). This method entails training a large-scale model, referred to as the "teacher" model, followed by transferring its knowledge to a smaller "student" model. The conventional distillation methods mainly concentrate on classification tasks, endeavoring to make the probability distribution of the student model's output as similar as possible to that of the teacher model. Recently, various distillation methods have been proposed for representation models (Tian et al., 2019; Wu et al., 2021; Zhao et al., 2022). The primary objective of these methods is to ensure that the student model's representation closely resembles that of the teacher model. This is typically achieved through the use of a learning objective such as mean squared error (MSE) or contrastive learning loss.

For an input text $X$, the final representations of the teacher model and student model are denoted as $T$ and $S$, respectively. From an information

---

[1]The source code is publicly available at https://github.com/Alibaba-NLP/IBKD.

theory perspective, the conventional distillation learning process of continuously approximating $T$ and $S$ through optimization objectives such as MSE or contrastive learning can be regarded as maximizing the mutual information (MI) between $T$ and $S$, which is denoted as $I(T, S)$. However, according to the Information Bottleneck (IB) principle (Tishby and Zaslavsky, 2015), simply maximizing $I(T, S)$ is prone to over-fitting. According to the IB principle, when given an input $X$ and its corresponding label $Y$, our goal is to learn a low-dimensional representation $Z$ that is informative for predicting $Y$ while minimizing the presence of irrelevant information. The IB method represents this information compressing process as maximizing the mutual information between $Z$ and $Y$ while minimizing the mutual information between $Z$ and $X$. This approach ensures that the most useful information is preserved, while redundant information is discarded. In text representation distillation, the representations of the teacher and student models, $T$ and $S$, can be equivalently considered as the label $Y$ and the compressed representation $Z$.

Therefore, we propose a new text representation distillation method based on the IB principle, called IBKD. It aims to maximize the mutual information between $S$ and $T$ while minimizing the mutual information between $S$ and $X$. As shown in Figure 1, compared to the conventional distillation methods (left), IBKD can effectively filters out task-irrelevant information and retains only the task-relevant information (right), thus improving the generalization of the student model's representation. Further, to address the issue of excessive computational effort in directly calculating mutual information (Alemi et al., 2017a), we introduce different methods to approximate the upper and lower bounds of mutual information. Specifically, we use contrastive learning loss (Sordoni et al., 2021) to estimate the lower bound of mutual information and the Hilbert-Schmidt independence criterion (HSIC) (Gretton et al., 2005) is used to estimate the upper bound.

Moreover, we found that two-stage distillation can significantly improve the performance of the student model. Concretely, the first stage distillation based on large-scale unsupervised data allows the student model to acquire basic text representation characteristics, while the second stage of distillation based on supervised data can further strengthen the representation ability of the student

model. Importantly, both two stages of distillation can be efficiently performed within the same framework. To demonstrate the effectiveness of our proposed method, we conduct experiments on two main downstream tasks of text representation: the semantic textual similarity (STS) task and the dense retrieval (DR) task. Our experimental results have shown that our approach significantly outperforms other methods.

Briefly, our main contributions are as follows: 1) Drawing on the IB principle, we propose a new IBKD method for text representation distillation. 2) We introduce the Contrastive learning loss and HSIC method to reduce the computational cost of mutual information. 3) We verify the effectiveness of IBKD on multiple benchmark datasets for two different down-streaming tasks.

## 2 Related Work

**Knowledge Distillation for text representation** Pretrained Language Models (PLMs) have demonstrated remarkable success in the field of text representation (Reimers and Gurevych, 2019; Karpukhin et al., 2020). Recent research has been devoted to enhancing the performance of PLM-based models through specific pretraining tasks (Gao and Callan, 2022; Wu et al., 2022; Long et al., 2022b; Shen et al., 2022), contrastive learning (Gao et al., 2021; Karpukhin et al., 2020), and hard negative mining (Xiong et al., 2021; Tabassum et al., 2022). However, these methods are primarily designed for large-scale PLMs, and their direct application to small-scale models often leads to a significant performance decline (Zhao et al., 2022).

Knowledge distillation, initially introduced by Hinton et al. (2015), is a technique employed to convert large, intricate models into smaller, more efficient models while preserving a high level of generalization power. Traditional knowledge distillation methods typically utilize a KL divergence-based loss to align the output logits of the "teacher" model and the "student" model (Zagoruyko and Komodakis, 2017; Sun et al., 2020).

Recently, new approaches have been developed specifically for representation-based models. For instance, the Contrastive Representation Distillation (CRD) method proposed by Tian et al. (2019) adopts a contrastive objective to match the representations of the teacher and student models. The HPD method (Zhao et al., 2022) aims to make the student model's representation similar to a com-

pressed representation of the teacher model, thus reducing both the model size and the dimensionality of the final output. The DistilCSE method (Wu et al., 2021) is a two-stage framework that first distills the student model using the teacher model on a large dataset of unlabeled data and subsequently fine-tunes the student model on labeled data.

**Information Bottleneck Principle** The information bottleneck (IB) principle (Tishby and Zaslavsky, 2015) refers to the tradeoff that exists in the hidden representation between the necessary information required for predicting the output and the information that is retained about the input. It has been applied in the study of deep learning dynamics (Saxe et al., 2018; Goldfeld et al., 2019), resulting in the creation of more interpretable and disentangled representations of data (Bao, 2021; Jeon et al., 2021). In addition, it has served as a training objective in recent works (Belghazi et al., 2018; Paranjape et al., 2020).

However, a significant challenge in the context of Information Bottleneck (IB) is the estimation of the joint distribution of two random variables and the calculation of the entropy of a random variable. In response to this challenge, the Variational Information Bottleneck (VIB) (Alemi et al., 2017b) method employs a variation approximation of the original IB, while the HSIC-Bottleneck (Ma et al., 2020) method replaces mutual information terms with the Hilbert-Schmidt Independence Criterion (HSIC) (Gretton et al., 2005) to assess the independence of two random variables. To the best of our knowledge, our research represents the first application of IB as a training objective in the knowledge distillation area.

## 3 Method

The objective of distillation is to transfer knowledge from a well-trained teacher model ($f^t$) to a student model ($f^s$). For input $X$, the representations of teacher and student are denoted as $T$ and $S$ respectively. Referring to the IB principle, we aims to maximize the mutual information between $S$ and $T$ and minimize the mutual information between $S$ and the original input $X$. Formally, the learning objective can be formulated as:

$$\mathcal{L}_{\text{IKBD}} = -I(S,T) + \beta \times I(X,S) \quad (1)$$

where $I(\cdot, \cdot)$ denotes the mutual information between two random variables and $\beta \geq 0$ is a hyperparameter controls the tradeoff between two parts.

However, directly optimizing $\mathcal{L}_{\text{IKBD}}$ is intractable, especially when $X$, $S$, $T$ are high dimensional random variables with infinite support (Alemi et al., 2017a). Consequently, we resort to estimating a lower bound of $I(S,T)$ and an approximation of $I(S,X)$ instead.

In the following subsections, we will first discuss the application of the contrastive learning loss and HSIC method in approximating the Information Bottleneck. Subsequently, we will explain how we apply them in our two-stage distillation process.

### 3.1 Lower Bound of $\text{I}(\text{S}, \text{T})$

To maximize the mutual information $I(S,T)$, we utilize the InfoNCE loss (Liu et al., 2022) as it has been demonstrated to be a lower bound for mutual information (Sordoni et al., 2021). The InfoNCE loss is defined as:

$$L_{nce} = \log \frac{P(S,T)}{P(S,T) + \mathbb{E}_S P(S,T)} \quad (2)$$

$$\approx \log \left( 1 + \frac{MP(S)P(T)}{P(S,T)} \right)$$

$$\leq -\log(M+1) + \log \frac{P(S,T)}{P(S)P(T)}$$

$$= -\log(M+1) + I(S,T), \quad (3)$$

where $P(S,T)$ represents the joint distribution of $S$ and $T$, and $P(S)$ and $P(T)$ represent the marginal distributions of $S$ and $T$ respectively. $M$ denotes the number of negative samples. Based on the above equation, it can be inferred that:

$$I(S,T) \geq L_{nce} + \log(M+1). \quad (4)$$

Thus, $L_{nce}$ can be treated as a lower bound for $I(S,T)$, and its tightness increases as $M$ grows.

In practice, we establish a connection between $P(s_i, t_i)$ and $sim(s_i, t_i) = \exp(\frac{s_i^T W t_i}{\tau})$, where $\tau$ represents the temperature, $W$ is a learnable matrix used to align the dimensions of $S$ and $T$, and $s_i$ and $t_i$ are instances of $S$ and $T$, respectively. Hence, we have:

$$L_{nce} = \log \frac{P(S,T)}{P(S,T) + \mathbb{E}_S P(S,T)} \quad (5)$$

$$= -\mathbb{E}_{s_i} \left[ \log \frac{sim(s_i, t_i)}{sim(s_i, t_i) + \sum_j^n sim(s_i, t_j)} \right]$$

$$\approx -\frac{1}{n} \sum_{x_i}^X \log \frac{sim(s_i, t_j)}{sim(s_i, t_i) + \sum_j^n sim(s_i, t_j)},$$

where $n$ denotes the batch size, and we employ other samples within the same batch as negative samples. In this particular setting, the number of negative samples $M$ is equal to $n$.

## 3.2 Approximation of $\mathbf{I}(\mathbf{X}, \mathbf{S})$

To minimize the mutual information between $X$ and $S$. We introduce Hilbert-Schmidt Independence Criterion (HSIC) (Gretton et al., 2005) here as its approximation. HSIC is a statistical method used to measure the independence between two random variables:

$$HSIC(X, S) = ||C_{XS}||^2_{hs},$$

where $|| \cdot ||^2_{hs}$ denotes the Hilbert-Schmidt norm (Gretton et al., 2005), and $C_{XS}$ is the cross-covariance operators between the Reproducing Kernel Hilbert Spaces (RKHSs) (Berlinet and Thomas-Agnan, 2004) of $X$ and $S$.

Previous studies (Ma et al., 2020) have proven that a lower $HSIC(X, S)$ indicates that $P(X)$ and $P(S)$ are more independent, and $HSIC(X, S) = 0$ if and only if $I(X, S) = 0$. So minimizing $HSIC(X, S)$ is equivalent to minimizing $I(X, S)$. Let $D = \{(x_1, s_1), (x_2, s_2), \ldots, (x_l, s_l)\}$ contains $l$ samples i.i.d drawn from $P(S, X)$, Gretton et al. (2005) proposed an empirical estimation of HSIC based on $D$:

$$\widehat{HSIC(X, S)} := \frac{1}{l^2} tr(K_X H K_S H), \quad (6)$$

the centering matrix is denoted as $H = I - \frac{1}{n}\mathbf{1}\mathbf{1}^T$, where $tr$ represents the trace of a matrix. $K_X$ and $K_S$ are kernel Gram matrices (Ham et al., 2004) of $X$ and $S$ respectively, where $K_{X_{ij}} = k(x_i, x_j)$ and $k(\cdot, \cdot)$ denotes a kernel function. In this paper, we use the commonly used radial basis function (RBF) kernel (Vert et al., 2004) by experiments

$$K(x_i, x_j) = \exp(\gamma||x_i - x_j||), \quad (7)$$

where $\gamma$ is a hyperparameter. It is worth highlighting that the above equation relies solely on positively paired samples to construct the kernel matrices $K_X$ and $K_Y$, which means that the calculation process is notably more efficient than directly estimating mutual information.

Although it is feasible to replace the InfoNCE loss with HSIC to maximize $I(S, T)$, previous research (Tschannen et al., 2020) has demonstrated that using the InfoNCE loss often leads to better

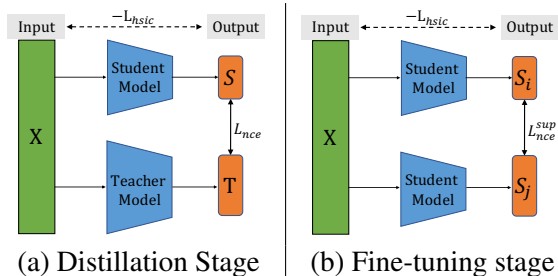

(a) Distillation Stage | (b) Fine-tuning stage

Figure 2: An illustration of the IBKD training process, with the left depicting the distillation stage and the right depicting the fine-tuning stage, respectively.

performance than directly maximizing mutual information. In our own experiments, we have also observed that the InfoNCE method tends to produce superior results in practice.

## 3.3 Two Stage Training

To fully utilize both unsupervised and supervised data, the training process of the student model consists of two stages, as outlined in prior research (Wu et al., 2021). In the first stage, referred to as the distillation stage, we utilize a substantial amount of unlabeled data to train the student model with the following loss function:

$$L_{IB} = -(L_{nce}(S, T) - \beta_1 \widehat{HSIC(X, S)}). \quad (8)$$

during this stage of training, the student model can acquire the text representation characteristics of the teacher model, effectively imbuing the former with the latter's features.

In the second stage, known as the fine-tuning stage, we subject the student model to further fine-tuning using labeled data. This process helps to mitigate any potential bias introduced by the use of unlabeled data in the previous stage. Importantly, the labeled data used in this stage can be identical to that which was used to train the teacher model.

Considering a labeled dataset consisting of multiple instances, where each instance contains an anchor sample $x_i$, a positive sample $x_i^+$, and a set of $K$ negative samples: $\{x_{i1}^-, x_{i2}^-, \ldots, x_{ik}^-\}$. In the fine-tuning stage, we continue to utilize a combination of the HSIC method and contrastive loss to prevent over-fitting:

$$L_{IB}^{sup} = -(L_{nce}^{sup} - \beta_2 \widehat{HSIC(X, S)}). \quad (9)$$

where $L_{nce}^{sup}$ is a supervised InfoNCE loss:

$$\frac{1}{l} \sum_i^l \log \frac{exp(\frac{s_i^T s_i^+}{\tau})}{\sum_j^l exp(\frac{s_i^T s_j^+}{\tau}) + \sum_k^K exp(\frac{s_i^T s_{ik}^-}{\tau})}.$$
(10)

An overview of the whole training process of IBKD is shown in Figure 2. After the fine-tuning stage, the final student model is obtained and can be utilized for downstream tasks.

### 3.4 Dimension Reduction

In many downstream applications of text representation models, such as information retrieval, the dimension of the text representation model has a direct impact on storage costs and search latency. To reduce the final dimension of the student model's representation, during the fine-tuning stage, we add a projection layer after getting $S$:

$$S' = W_{proj}S,$$
(11)

where $W_{proj} \in R^{d \times d'}$, $d$ and $d'$ represent the dimensions of the original student model and the final dimension, respectively. During inference, we will utilize $S'$ as the final representation.

## 4 Experiment

### 4.1 Experiment Setup

We conduct experiments on two tasks: semantic textual similarity (STS) and dense retrieval (DR). The STS task aims to measure the semantic similarity between two sentences, for this task, we evaluate seven standard datasets: STS12-16 (Agirre et al., 2012, 2013, 2014, 2015, 2016), STS-B (Cer et al., 2017) and SICK-R (Marelli et al., 2014). Following previous work, we use the SentEval toolkit (Conneau and Kiela, 2018) to do the evaluation and use Spearman's rank correlation as the performance metric.

The DR task aims to retrieval relevant passages of the given query, for this task we do experiments on the MS MARCO Passage Ranking Dataset (Nguyen et al., 2016). We use MRR@10 and Recall@1000 as the evaluation metrics.

### 4.2 Baseline Models

We select two pretrained models of varying sizes, namely TinyBERT-L4 (Jiao et al., 2020) [2] and

[2]https://huggingface.co/nreimers/TinyBERT_L-4_H-312_v2

MiniLM-L6 (Wang et al., 2020) [3], as student models following previous works (Wu et al., 2021; Zhao et al., 2022). For the STS task, we employed the state-of-the-art model SimCSE-RoBERTa$_{large}$ [4] as the teacher model, while for the DR task, we utilized CoCondenser [5] as the teacher model. Unless explicitly stated, in the following, the term "Model-IBKD" represents a model that has undergone the distillation training stage, while "Model-IBKD$_{ft}$" refers to the model after the fine-tuning stage's training.

Our baseline models include two types: the first involves directly training different-sized pretrained models on the supervised dataset, and the second involves using previous state-of-the-art knowledge distillation methods for representation models, such as the traditional MSE loss based representation distillation method (Kim and Rush, 2016), the HPD (Zhao et al., 2022) method and the CRD method (Tian et al., 2019). In addition, we also fine-tune the HPD model (HPD$_{ft}$) using the training data and contrastive learning loss for each task based on their public model [6] as a baseline model to estimate the impact of the fine-tuning stage.

|  | STS | STS$_{ft}$ | DR | DR$_{ft}$ |
|---|---|---|---|---|
| learning rate | 1e-4 | 3e-5 | 1e-4 | 1e-5 |
| batch size | 256 | 256 | 128 | 128 |
| epoch | 10 | 3 | 3 | 3 |
| $\tau$ | 0.1 | 0.05 | 0.1 | 0.05 |

Table 1: Hyparameters for STS and DR tasks in different stages. The subscript "ft" indicates the fine-tuning stage. We use the same hyparameters for both TinyBERT-L4 and MiniLM-L6 models.

### 4.3 Implementation Details

**Training Data** For the STS task, we utilized the same dataset as described in Zhao et al. (2022) for the first stage training. This dataset comprises the original Natural Language Inference (NLI) dataset along with additional data generated by applying WordNet substitution and back translation to each instance of the NLI dataset. In the fine-tuning stage, we used the "entailment" pairs from the original

[3]https://huggingface.co/nreimers/MiniLM-L6-H384-uncased
[4]https://huggingface.co/princeton-nlp/sup-simcse-roberta-large
[5]https://huggingface.co/Luyu/co-condenser-marco-retriever
[6]https://huggingface.co/Xuandong

| Model | STS12 | STS13 | STS14 | STS15 | STS16 | STS-B | SICK-R | Avg | Params | Dimension |
|---|---|---|---|---|---|---|---|---|---|---|
| SimCSE-RoBERTa$_{base}$ | 76.53 | 85.21 | 80.95 | 86.03 | 82.57 | 85.83 | 80.50 | 82.52 | 110M | 768 |
| SimCSE-RoBERTa$_{large}$♠ | 77.46 | 87.27 | 82.36 | 86.66 | 83.93 | 86.70 | 81.95 | 83.76 | 330M | 1024 |
| SimCSE-TinyBERT | 73.02 | 80.71 | 76.89 | 83.01 | 78.57 | 81.10 | 78.19 | 78.78 | 14M | 312 |
| TinyBERT-MSE | 73.07 | 81.83 | 77.92 | 84.49 | 80.35 | 81.69 | 79.10 | 79.78 | 14M | 312 |
| TinyBERT-HPD | 74.29 | 83.05 | 78.80 | 84.62 | 81.17 | 84.36 | **80.83** | 81.02 | 14M | 128 |
| TinyBERT-HPD$_{ft}$ | 75.17 | 84.10 | 79.97 | 85.44 | 82.17 | 85.52 | 80.65 | 81.60 | 14M | 128 |
| TinyBERT-CRD | 74.56 | 83.26 | 78.71 | 84.86 | 80.72 | 82.11 | 79.55 | 80.54 | 14M | 312 |
| TinyBERT-IBKD | 74.73 | 83.56 | 78.97 | 84.97 | 81.68 | 84.37 | 80.52 | 81.69 | 14M | 312 |
| TinyBERT-IBKD$_{ft}$ | **76.14** | **84.45** | **80.19** | 85.54 | **82.51** | **85.09** | 80.18 | **82.01** | 14M | 312 |
| TinyBERT-IBKD-128 | 74.10 | 83.59 | 79.38 | 85.65 | 81.53 | 83.87 | 79.73 | 81.12 | 14M | 128 |
| SimCSE-MiniLM | 70.34 | 78.59 | 75.08 | 81.10 | 77.74 | 79.39 | 77.85 | 77.16 | 23M | 384 |
| MiniLM-MSE | 73.75 | 81.42 | 77.72 | 83.58 | 78.99 | 81.19 | 78.48 | 79.30 | 23M | 384 |
| MiniLM-HPD | 74.94 | 84.52 | 80.25 | 84.87 | 81.90 | 84.98 | 81.15 | 81.80 | 23M | 128 |
| MiniLM-HPD$_{ft}$ | 76.03 | 84.71 | 80.45 | 85.53 | 82.07 | 85.33 | 80.01 | 82.05 | 23M | 128 |
| MiniLM-CRD | 74.79 | 84.19 | 78.98 | 84.70 | 80.65 | 82.71 | 79.91 | 81.30 | 23M | 384 |
| MiniLM-IBKD | 75.57 | 85.41 | 80.27 | 84.99 | 82.46 | 84.78 | 80.48 | 82.01 | 23M | 384 |
| MiniLM-IBKD$_{ft}$ | **76.77** | **86.13** | **81.03** | **85.66** | **82.81** | **86.14** | **81.25** | **82.69** | 23M | 384 |
| MiniLM-IBKD-128 | 76.34 | 85.38 | 81.32 | 85.34 | 81.87 | 85.14 | 80.67 | 82.29 | 23M | 128 |

Table 2: Results on the STS datasets. The teacher model is marked with ♠. Dimension denotes the output's dimension of the text representation. We bold the best performance of each student's backbone. The results for SimCSE and HPD methods are taken from their respective original papers, while all other results were reproduced by us. TinyBERT-IBKD-128 and MiniLM-IBKD-128 denote the IBKD models fine-tuned with the dimension reduction method, and the final dimension set to 128. The results of IBKD models are statistically significant difference ($p < 0.01$) compared to other models.

NLI dataset as positive pairs and the "contradiction" pairs as negative pairs.

For the DR task, we utilized all passages and training queries provided by the MS MARCO Passage Dataset during the initial training stage. We then conducted fine-tuning using the labeled data from the same dataset. In this task, the negatives were acquired by leveraging the CoCondenser model. More details are reported in the Appendix.

**Optimizing Setup** The values of hyperparameters are listed in Table 1. We kept the value of $\gamma$ at 0.5, $\beta_1$ at 1.0, and $\beta_2$ at 0.5 for all experiments. All hyperparameters are selected through grid search. The search range for each hyperparameters are listed in the Appendix. For the fine-tuning stage, we select 8 hard negatives for each query. We used Adam for optimization. Our code was implemented in Python 3.7, using Pytorch 1.8 and Transformers 2.10. All experiments were run on a single 32G NVIDIA V100 GPU. For the DR task, we constructed the index and performed ANN search using the FAISS toolkit (Johnson et al., 2021).

### 4.4 Experiment Results on STS

From the STS results in Table 2, we observe that: 1) Our method outperforms previous methods using the same student model. For instance,

the MiniLM-IBKD model delivered a Spearman's rank correlation performance of $98.72\%$ while employing just $6.9\%$ of the parameters utilized by SimCSE-RoBERTa$_{large}$. Remarkably, it outperforms SimCSE-RoBERTa$_{base}$, which has 4.7 times more parameters. 2) After the distillation training stage, IBKD has already outperformed previous distillation methods, and the subsequent fine-tuning stage yields additional performance gains. For instance, fine-tuning led to a $0.40\%$ improvement for the TinyBERT model and a $0.68\%$ improvement for the MiniLM student model. 3) Although applying the dimension reduction method reduces performance, it remains competitive performance with previous state-of-the-art results.

### 4.5 Experiment Results on DR

Table 3 shows the results on the DR task. We can find that: 1) IBKD effectively reduces the disparity between the student and teacher models compared to previous methods. For example, the MiniLM-IBKD model attains a $97.9\%$ MRR@10 performance with only $4.2\%$ of parameters compared to the teacher model CoCondenser. The smaller parameters make it $4.6$ times faster than the teacher model and require only $50\%$ of memory to store the embeddings. 2) The fine-tuning stage has a more significant impact on performance in the DR

| Model | MRR@10 | Recall@1000 | Dimension | Params | Speed | Memory |
|---|---|---|---|---|---|---|
| CoCondenser ♠ | 38.21 | 98.40 | 768 | 110M | 500 | 26G |
| TinyBERT-sup | 28.64 | 89.97 | 312 | 14M | 3000 | 11G |
| TinyBERT-MSE | 25.98 | 90.11 | 312 | 14M | 3000 | 11G |
| TinyBERT-CRD | 27.40 | 92.54 | 312 | 14M | 3000 | 11G |
| TinyBERT-HPD | 27.77 | 92.99 | 128 | 14M | 3000 | 4.4G |
| TinyBERT-HPD$_{ft}$ | 34.93 | 96.04 | 128 | 14M | 3000 | 4.4G |
| TinyBERT-IBKD | 28.88 | 90.02 | 312 | 14M | 3000 | 11G |
| TinyBERT-IBKD$_{ft}$ | **37.32** | **97.46** | 312 | 14M | 3000 | 11G |
| TinyBERT-IBKD-128 | 35.57 | 96.70 | 128 | 14M | 3000 | 4.4G |
| MiniLM-sup | 30.51 | 94.32 | 384 | 23M | 2300 | 13G |
| MiniLM-MSE | 28.12 | 93.01 | 384 | 23M | 2300 | 13G |
| MiniLM-CRD | 28.79 | 93.12 | 384 | 23M | 2300 | 13G |
| MiniLM-HPD | 29.79 | 93.98 | 128 | 23M | 2300 | 4.4G |
| MiniLM-HPD$_{ft}$ | 36.53 | 96.70 | 128 | 23M | 2300 | 4.4G |
| MiniLM-IBKD | 30.31 | 93.66 | 384 | 23M | 2300 | 13G |
| MiniLM-IBKD$_{ft}$ | **37.49** | **97.81** | 384 | 23M | 2300 | 13G |
| MiniLM-IBKD-128 | 36.32 | 97.01 | 128 | 23M | 2300 | 4.4G |

Table 3: Performance of different models on the MS MARCO Passage Ranking Dataset. The teacher model is marked with ♠. The results of IBKD are statistically significant difference ($p < 0.01$) compared to other models. We bold the best performance of each student backbone. All results, except for CoCondenser, are from our implementation. The speed denotes the number of sentences encoded by the model per second using one V100.

task. Specifically, when using the MiniLM-L6-HPD model and the MiniLM-IBKD model, the fine-tuning stage resulted in a $5.74\%$ increase and $7.0\%$ increase in MRR@10, respectively. This can be attributed to the fact that the DR task is an asymmetric matching task, and labeled data plays a crucial role in guiding the model to accurately derive the semantic association between the query and the document. 3) Applying the dimension reduction method has been observed to decrease model performance by approximately $1\%$. However, this tradeoff is offset by the significant advantage of saving up to $60\%$-$70\%$ of memory costs.

To ensure our method is robust across teacher models, we conducted experiments for each task using an alternative teacher model. The corresponding results are reported in the Appendix.

## 4.6 Ablation Study

**Impact of the HSIC loss and Kernel Methods** In this study, we aim to evaluate the impact of the HSIC loss and various Gram Kernel methods on HSIC. Specifically, we compare the original IBKD model's performance with a distillation-based approach that only employs the contrastive learning loss in the distillation and fine-tuning stages (referred to as CKD here). Moreover, we examine the

| Model | STS | DR |
|---|---|---|
| TinyBERT-L4-IBKD$_{ft}$ | 82.01 | 37.32 |
| TinyBERT-L4-CKD | 81.37 | 35.73 |
| w IMQ kernel | 81.94 | 36.83 |
| w linear kernel | 81.26 | 36.17 |

Table 4: Experiment results on the STS and DR task with different kernel method. We report the Average Spearman's rank correlation coefficient of the seven datasets as the performance metric for the STS task and MRR@10 for the DR task respectively.

effectiveness of three types of Gram Kernel methods, namely linear, RBF, and inverse multiquadric (IMQ) (Javaran and Khaji, 2012). Our results, presented in Table 4, reveal that the implementation of HSIC loss can significantly enhance the performance of both STS and DR tasks. Additionally, we observe that the linear kernel leads to decreased performance, while the non-linear kernels (RBF and IMQ) exhibit comparable performance levels.

**Impact of the $\beta$ value** We conducted an analysis to assess the impact of $\beta$ on the performance of IBKD. The results of the STS task are presented in Figure 3, depicting both the distillation stage (in blue) and the fine-tuning stage (in orange).

In the distillation stage, we observed a notable

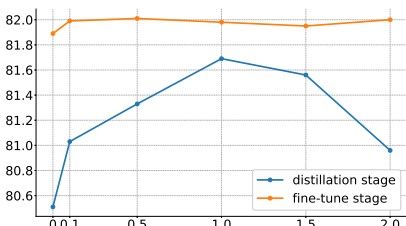

Figure 3: STS results with different $\beta$ for IBKD in the distillation (blue) and fine-tuning (orange) stages.

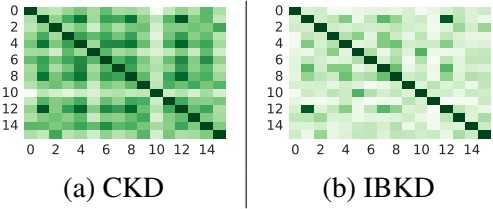

(a) CKD    (b) IBKD

Figure 4: The covariance matrix of the student model's representation. The darker color denotes a large correlation value. Due to space limitation, we only report the results for the first 16 dimensions.

improvement in the performance as $\beta$ increased. However, beyond a certain threshold, the performance began to decline. Our findings indicate that achieving an optimal balance between maximizing $I(S, T)$ and minimizing $I(S, X)$ can significantly enhance the student model's ability for downstream tasks. Regarding the fine-tuning stage, we noticed that the model's performance remained relatively consistent across different values of $\beta$. This observation can likely be attributed to the fact that the model's representations have already undergone training to exhibit low mutual information with the input during the preceding distillation stage.

## 4.7 Analysis

In this section, we aim to examine the effect of minimizing mutual information between the student model and its input. Therefore, we conduct several experiments using the TinyBERT-L4 model as the student model while maintaining consistent hyperparameters with the primary experiment.

**Feature correlation** To analyze the feature correlation of the CKD and IBKD models, we randomly select 10,000 sentences from the Natural Language Inference (NLI) dataset and calculate the covariance matrix for each dimension. As depicted in Figure 4 , the representations generated by the IBKD model exhibit low covariance across all dimensions, indicating that IBKD facilitates the learning of a more disentangled representation.

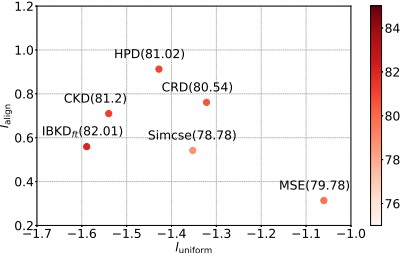

Figure 5: Alignment and Uniform metric of different Knowledge Distillation method. The darker color denotes a better performance.

**Alignment and Uniform** We evaluate the quality of embedding of different distillation methods through two widely used metrics: alignment and uniformity Wang and Isola (2020). Alignment measures the expected distance between positive pairs:

$$L_{\text{align}} := \mathbb{E}_{(x,y) \sim P_{pos}(x,y)}[||f(x) - f(y)||_2^2]. \quad (12)$$

On the other hand, uniformity measures the expected distances between embeddings of two random examples:

$$L_{\text{uniform}} := \log \mathbb{E}_{(x,y) \sim P_{data}(x,y)}[e^{-2||f(x)-f(y)||_2^2}]. \quad (13)$$

We plot the distribution of the "uniformity-alignment" map for different representation models across different distillation methods in Figure 5. The uniformity and alignment are calculated on the STS-B dataset. For both uniformity and alignment, lower values represent better performance. We observe that our IBKD method achieves a better trade-off between uniformity and alignment compared with other distilled models.

## 5 Conclusion

In this paper, we present a new approach for distilling knowledge in text representation called IBKD. Our technique is created to address the performance gap between large-scale pre-trained models and smaller ones. Drawing inspiration from the Information Bottleneck principle, IBKD selectively retains crucial information for the student model while discarding irrelevant information. This assists the student model in avoiding over-fitting and achieving a more disentangled representation. Through empirical experiments conducted on two text representation tasks, we demonstrate the effectiveness of IBKD in terms of accuracy and efficiency. These results establish IBKD as a promising technique for real-world applications.

# 6 Limitation and Risk

The IBKD method has two primary limitations. Firstly, it requires that the teacher model be a representation model, which limits the use of other architectures, such as cross-encoder models that take a pair of texts as input and output their semantic similarity score. Secondly, the fine-tuning stage need labeled data which may not be avaliabel in certain situation, even though these data can be the same data as that used to train the teacher model. Additionally, our model may perpetuate biases, lack transparency, pose security and privacy risks, similar to other pre-trained models.

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

## A Appendix-A

### A.1 Statistics of the Dataset

The NLI dataset consists of 1 million pairs and 1,196,755 sentences. After data augmentation, the dataset includes 3,590,265 sentences. The MS MARCO dataset includes 502,939 training queries, 8,841,823 documents, and 6,980 test queries.

### A.2 Hyparameter search range

Table 5 list the range of each hyparameter we used. We select the final hyparameter on the STS-B dev set and MS MARCO dev set for the STS task and DR task respectively.

Table 5: Hyparameters search range.

| Hyparameter | range |
|---|---|
| learning rate | 1e-5, 3e-5, 1e-4, 1e-4 |
| batch size | 64, 128, 256 |
| epoch | 3, 5, 10 |
| $\tau$ | 0.01, 0.05, 0.1, 0.2, 0.5 |
| $\gamma$ | 0.01,0.1,0.5,1.0 |
| $\beta_1$ | 0.1, 0.5, 1.0, 2.0 |
| $\beta_2$ | 0.1, 0.5, 1.0, 2.0 |

### A.3 Experiment results with additional teacher model

In this section, we present experimental results utilizing additional teacher models for the STS and DR tasks. For the STS task, we selected SimCSE-BERT$_{large}$ (Gao et al., 2021) as our teacher model, while for the DR task, we chose RetroMAE (Xiao et al., 2022). As a baseline, we employed the HPD method. The corresponding results are provided in Table 6 for the STS task and Table 7 for the DR task.

| Model | Avg | Params | Dimension |
|---|---|---|---|
| SimCSE-BERT$_{large}$ | 82.21 | 330M | 1024 |
| TinyBERT-HPD | 80.24 | 14M | 128 |
| TinyBERT-HPD$ft$ | 80.88 | 14M | 128 |
| TinyBERT-IBKD | 81.02 | 14M | 312 |
| TinyBERT-IBKD$ft$ | 81.43 | 14M | 312 |

Table 6: Experiment resulst for the STS task with SimCSE-BERT-large as the teacher model.

The additional experiments above indicate that different teacher models of IBKD consistently demonstrate performance, effectively enabling the

| Model | MRR@10 | Recall@1000 | Params | Dimension |
|---|---|---|---|---|
| RetroMAE | 41.6 | 98.8 | 110M | 768 |
| TinyBERT-HPD | 25.38 | 92.55 | 14M | 128 |
| TinyBERT-HPD | 36.93 | 98.04 | 14M | 128 |
| TinyBERT-IBKD | 30.12 | 92.04 | 14M | 312 |
| TinyBERT-IBKD | 38.21 | 98.20 | 14M | 128 |

Table 7: Results of the DR task with Retromae as the teacher model.

student model to closely resemble the teacher model in comparison to other methods. This also confirms the generalizability of the IBKD approach.