# OpenReview forum: "Text Representation Distillation via Information Bottleneck Principle"
_EMNLP/2023/Conference — EMNLP 2023 Main_

### Official Review · Reviewer_gS4Y · 2023-08-03

**Soundness:** 3

**Excitement:**

3: Ambivalent: It has merits (e.g., it reports state-of-the-art results, the idea is nice), but there are key weaknesses (e.g., it describes incremental work), and it can significantly benefit from another round of revision. However, I won't object to accepting it if my co-reviewers champion it.

**Paper Topic And Main Contributions:**

This paper proposes a knowledge distillation method in text representation called IBKD that aims to address the performance gap between large-scale pre-trained models and smaller ones. Inspired from the Information Bottleneck (IB) principle, the method maximizes mutual information between the final representation of the teacher and student model and minimizes mutual information between the representation of the student model and input. The paper also introduces the contrastive learning loss and HSIC method to reduce the computational cost of mutual information.

**Questions For The Authors:**

A.	Why were STS and DR selected as downstream tasks to evaluate distillation performance?
B.	What data was used for unsupervised learning stage?
C.	It is not clear whether the teacher model was fine-tuned for the STS and DR tasks or not.


**Reasons To Accept:**

This method outperformed other distillation methods on two downstream tasks (STS, DR).
This paper showed that the contrastive learning loss and HSIC method were effective for performance improvement.


**Reasons To Reject:**

It is stated that “simply maximizing I(T,S) is prone to over-fitting. However, IBKD can effectively filter out task-irrelevant information and retains only the task-relevant information, thus improving the generalization of the student model’s representation”. However, there is no result showing that the proposed method can improve generalization ability.

The range of tasks used for evaluation is narrow. When assessing distillation performance, many papers use benchmark datasets like GLUE, which evaluate the model's ability in various aspects.


**Reproducibility:**

5: Could easily reproduce the results.

**Reviewer Confidence:**

4: Quite sure. I tried to check the important points carefully. It's unlikely, though conceivable, that I missed something that should affect my ratings.

**Typos Grammar Style And Presentation Improvements:**

Is it correct that the dimension of TinyBERT-IBKD-128 is 312? It seems like a typo.

---

> ### Author Rebuttal · Authors · 2023-08-28
>
> Dear reviewer,
> We are very grateful for your recognition of our efforts. We have carefully considered your thoughtful comments and we'd like to offer some clarifications as well as solutions that aim to address the concerns you raised.
>
> > It is stated that “simply maximizing I(T,S) is prone to over-fitting. However, IBKD can effectively filter out task-irrelevant information and retains only the task-relevant information, thus improving the generalization of the student model’s representation”. However, there is no result showing that the proposed method can improve generalization ability.
>
> It is important to clarify that in the task of text representation model distillation, the generalization ability of the student model is typically determined by the performance of the student model on the test dataset, rather than simply fitting the representation of the teacher model. Alternatively, we can view this as the student model maximizing its fit to the training dataset, as the teacher model is trained on this dataset and can be seen as another form of the training dataset.
>
> To validate our claims, we selected two widely recognized downstreaming tasks of text representation, namely senmatic text similarity (STS) and dense passage retrieval (DR). Our experimental results demonstrate that traditional model distillation methods, particularly those that directly approximate the teacher and student model representations using the Mean Squared Error (MSE) loss function, significantly degrade performance on the test set and undermine the generalization of the student model. However, the IBKD method proposed in this paper effectively mitigates this loss of generalization performance.
>
> In the revised version, we will provide a more detailed explanation to elucidate the improvement in model generalization performance.
>
> > The range of tasks used for evaluation is narrow. When assessing distillation performance, many papers use benchmark datasets like GLUE, which evaluate the model's ability in various aspects.
>
> > Why were STS and DR selected as downstream tasks to evaluate distillation performance?
>
> In this paper, our main objective was to develop a distillation method for the text representation model. Building upon previous work, we selected two downstream tasks that are most representative of text representation to evaluate the performance of our model. As for the GLUE benchmark you mentioned, it is commonly used by researchers to assess the overall performance of pre-trained language models. Typically, this evaluation process involves separate training for each individual task, which is not suitable for the context of our paper focusing on the representation model. However, it is worth noting that our proposed distillation method has the potential to be extended to the distillation process of the entire pre-training base model or other task models, rather than just limited to the distillation of the representation model. We leave further exploration for future studies.
>
>
> > What data was used for unsupervised learning stage
>
> The training data used for the unsupervised learning stage was discussed in Section 4.3 of the original paper. In the case of the STS task, we employed an augmented Natural Language Inference (NLI) dataset. This dataset comprises the original NLI dataset, supplemented with extra data created by applying WordNet substitution and back translation to every instance. As for the DR task, we utilized all passages and training queries from the MS MARCO Passage Dataset during the initial training stage.
>
> > It is not clear whether the teacher model was fine-tuned for the STS and DR tasks or not.
>
> The teacher model for each task undergoes fine-tuning using task-specific supervised data. The coCondenser model is trained on the MSMARCO dataset, while Simcse-Roberta is fine-tuned using the NLI dataset.
>
> We will clarify all these details above in the revised version and fix all typos.  Thanks again for your comments.

---

### Official Review · Reviewer_BBAJ · 2023-08-04

**Typos Grammar Style And Presentation Improvements:** N/A
**Soundness:** 4

**Excitement:**

4: Strong: This paper deepens the understanding of some phenomenon or lowers the barriers to an existing research direction.

**Missing References:**

Though the paper focuses on distillation via the mutual information technique, referencing some other works on BERT distillation would be beneficial:
1) BERT-of-Theseus: Compressing BERT by Progressive Module Replacing. ACL 2020.
2) DistilBERT, a distilled version of BERT: smaller, faster, cheaper and lighter.


**Paper Topic And Main Contributions:**

The paper optimizes Text Representation Distillation using the Information Bottleneck Principle. The work falls under computationally-aided linguistic analysis. The authors argue that traditional distillation methods based on mutual information retain some noise information from the teacher model, which is unnecessary for downstream tasks. Therefore, they adopt the Information Bottleneck strategy to constrain the mutual information between the student model representation and the input text to be as small as possible, while distilling information from the teacher model to the student model. Moreover, they propose a two-stage training method to optimize the model. The first stage is distillation learning, and the second stage is model fine-tuning on supervised data. The main contributions of this work are: 1) the application of the information bottleneck theory to language model distillation and the proposal of a two-stage training method to improve model performance; 2) thorough experimental validation of the model's effectiveness, along with the publication of detailed descriptions and key parameters.

**Questions For The Authors:**

A. My main question aligns with the Reasons to Reject, that is, how to use formula (7) to calculate the RBF kernel between two texts?

**Reasons To Accept:**

A. The model design is reasonable, including the use of the information bottleneck and the two-stage training method.
B. The effectiveness of the model is well demonstrated by extensive experiments, and the authors provide detailed parameter settings.


**Reasons To Reject:**

A. The process of estimating mutual information with HSIC is not detailed enough, especially on how to use formula (7) to calculate the RBF kernel between two texts?


**Reproducibility:**

4: Could mostly reproduce the results, but there may be some variation because of sample variance or minor variations in their interpretation of the protocol or method.

**Reviewer Confidence:**

4: Quite sure. I tried to check the important points carefully. It's unlikely, though conceivable, that I missed something that should affect my ratings.

---

> ### Author Rebuttal · Authors · 2023-08-28
>
> Dear Reviewer,
>
> Thank you for your insightful comments and suggestions. We have carefully reviewed your feedback and have made the following detailed responses:
>
> > The process of estimating mutual information with HSIC is not detailed enough, especially on how to use formula (7) to calculate the RBF kernel between two texts?
>
> We apologize for the insufficient explanation regarding the calculation process of HSIC. In the revised version, we will address this concern by providing a clearer explanation of the process.
>
> In our original paper, we introduced formula (6) as the method to calculate the HSIC between the input $X$ and student embedding $S$. To compute HSIC, it is necessary to calculate the kernel Gram matrices $K_x ∈ R^{n \times n}$ for $X$ and Ks ∈ R^{n x n} for S. Here, $K_{x_{ij}} = k(x_i, x_j)$ and $K_{s_{ij}} = k(s_i, s_j)$, where $x_i$ represents the input representation of the i-th text, and $s_i$ represents the output of the student model for the i-th text. The function $k(\cdot, \cdot )$ can be any kernel function. In this study, we utilize the RBF kernel, as described in formula (7), to calculate $k(x_i, x_j)$.
>
> > Though the paper focuses on distillation via the mutual information technique, referencing some other works on BERT distillation would be beneficial:  \
> > 1. BERT-of-Theseus: Compressing BERT by Progressive Module Replacing. ACL 2020. \
> > 2. DistilBERT, a distilled version of BERT: smaller, faster, cheaper and lighter.
>
> We will cite these papers you mentioned and discussed them in the related work section.
>
> Thanks again for your comments.

---

### Official Review · Reviewer_QEfC · 2023-08-04

**Soundness:** 5

**Excitement:**

4: Strong: This paper deepens the understanding of some phenomenon or lowers the barriers to an existing research direction.

**Paper Topic And Main Contributions:**

The paper proposes the IBKD method for making pre-trained language models (PLMs) more accessible by distilling them into smaller representation models. By implementing the Information Bottleneck (IB) principle, this method reduces information loss during the distillation process and concentrates on the most informative features of the PLMs. A standout aspect of this approach is the addition of a novel regularization term to counteract overfitting, as well as a well-devised implementation strategy for the IB principle. Experimental results underscore the method's effectiveness, demonstrating substantial performance improvements across a variety of text classification tasks.

**Questions For The Authors:**

Have you considered applying your method to other representation models beyond Roberta and CoCondenser to verify its generalizability?

**Reasons To Accept:**

- Good writing and easy to follow
- It is an inspiring idea to introduce the Information Bottleneck into the knowledge distillation
- The experiments conducted provide robust evidence for the effectiveness of the proposed method in knowledge distillation

**Reasons To Reject:**

- Limited Variety: The proposed method's application is primarily demonstrated on Roberta and CoCondenser. This limited variety in implementation may raise questions about the framework's generalizability. It's unclear whether the method would be as effective if applied to other representation models.
- Insufficient experiments: The experimentation conducted in the paper could be more comprehensive. The authors predominantly use a single teacher model for each dataset, which may not fully demonstrate the effectiveness of their method. Including at least one more teacher model in each dataset could provide a more robust validation of the method's efficacy.

**Reproducibility:**

4: Could mostly reproduce the results, but there may be some variation because of sample variance or minor variations in their interpretation of the protocol or method.

**Reviewer Confidence:**

3: Pretty sure, but there's a chance I missed something. Although I have a good feel for this area in general, I did not carefully check the paper's details, e.g., the math, experimental design, or novelty.

**Typos Grammar Style And Presentation Improvements:**

Figure 1 currently suggests that the teacher embeddings encompass all task-relevant information from the original input, which might oversimplify the knowledge distillation process. Authors are suggested to modify Figure 1 to illustrate partial overlap between the teacher embeddings and the task-relevant information, indicating the potential information loss from the teacher model.

---

> ### Author Rebuttal · Authors · 2023-08-28
>
> Dear Reviewer,
> We are very grateful for your evaluation of our paper and the time you've dedicated to providing your feedback. In the subsequent sections, we endeavor to respond to your comments and address the concerns you have raised.
>
>
> > Limited Variety: The proposed method's application is primarily demonstrated on Roberta and CoCondenser. This limited variety in implementation may raise questions about the framework's generalizability. It's unclear whether the method would be as effective if applied to other representation models.
>
> Given your concern about the generalization of our proposed method, we have chosen additional models as teacher models to validate the generalization ability of our method. It should be noted that, apart from replacing the teacher model, all other hyperparameters remain consistent with the experimental parameters in Table-1 of the original paper. Due to time constraints, we have only selected the HPD method as the baseline for validation.
>
> We employ the Simcse-BERT-large[1] model as the teacher model for the STS task. We use the term "Model-IBKD" to denote a model that has undergone the distillation training stage, and "Model-IBKD$_{ft}$" to denote the model after the fine-tuning stage's training. The results are presented below.
>
> | Model                 | STS12 | STS13 | STS14 | STS15 | STS16 | STS-B | SICK-R |  Avg  | Params | Dimension |
> |-----------------------|-------|-------|-------|-------|-------|-------|--------|-------|-------|-------|
> | Simcse-BERT-large     | 75.78 | 86.33 | 80.44 | 86.06 | 80.86 | 84.87 | 81.14  | 82.21 | 330M | 1024 |
> | TinyBERT-HPD          | 72.43 | 83.10 | 77.92 | 84.68 | 80.38 | 83.16 | 79.99  | 80.24 | 14M | 128 |
> | TinyBERT-HPD$_{ft}$       | 73.00 | 83.25 | 79.08 | 84.74 | 81.31 | 83.87 | 80.89  | 80.88 | 14M | 128 |
> | TinyBERT-IBKD         | 74.11 | 83.13 | 78.98 | 85.10 | 81.31 | 84.01 | 80.51  | 81.02 | 14M | 312 |
> | TinyBERT-IBKD$_{ft}$      | 74.48 | 83.39 | 79.40 | 85.28 | 81.80 | 84.44 | 81.21  | 81.43 | 14M | 312 |
>
>
> For the DR task, we choose another competitve pretrained BERT model RetroMAE[2] as the teacher model. The results are shown below:
>
> | Model             | MRR@10 | Recall@1000 | Params | Dimension |
> |-------------------|--------|-------------|------------|------------|
> | RetroMAE          | 41.6   | 98.8        | 110M | 768 |
> | TinyBERT-HPD      | 25.38  | 92.55       | 14M  | 128  |
> | TinyBERT-HPD$_{ft}$   | 36.93  | 98.04       | 14M  | 128 |
> | TinyBERT-IBKD | 30.12  | 92.04       | 14M | 312 |
> | TinyBERT-IBKD$_{ft}$  | 38.21  | 98.20       | 14M  | 128 |
>
> The additional experiments above indicate that different teacher models of IBKD consistently demonstrate performance, effectively enabling the student model to closely resemble the teacher model in comparison to other methods. This also confirms the generalizability of the IBKD approach.
>
> > Insufficient experiments: The experimentation conducted in the paper could be more comprehensive. The authors predominantly use a single teacher model for each dataset, which may not fully demonstrate the effectiveness of their method. Including at least one more teacher model in each dataset could provide a more robust validation of the method's efficacy.
>
> We will add all the above experiment results in the revised version and discussed with them in detail.
>
> > Figure 1 currently suggests that the teacher embeddings encompass all task-relevant information from the original input, which might oversimplify the knowledge distillation process. Authors are suggested to modify Figure 1 to illustrate partial overlap between the teacher embeddings and the task-relevant information, indicating the potential information loss from the teacher model.
>
>
> Thank you for your valuable suggestion. The main aim of Figure 1 is to emphasize the problem of over-fitting in the conventional distillation method. To effectively convey this message, we have assumed an ideal scenario where the teacher model incorporates all the required task-related information. However, we recognize that this simplification may have led to some misunderstandings.
> To address this concern, we will revise Figure 1 to illustrate a partial overlap between the teacher embeddings and the task-relevant information. This adjustment will provide a more precise representation of the real situation and reduce any potential confusion.
>
> Thanks again for your comments.
>
> **Reference**:
>
> [1] Simple Contrastive Learning of Sentence Embeddings
>
> [2] RetroMAE: Pre-Training Retrieval-oriented Language Models Via Masked Auto-Encoder

---

### Meta-Review · Area_Chair_f53V · 2023-09-20

**Recommendation:** 4

**Metareview:**

This paper proposes the IBKD method in order to make pre-trained language models (PLMs) more accessible by distilling them into smaller representation models. The topic studied is an interesting topic and the proposed method based on information bottleneck is meaningful. Reviewers have concerns on the insufficient experiments or missing details. During the rebuttal period, the authors have replied to these concerns with more details and experimental results, which lead to overall positive scores from reviewers.

---

### Decision · Program_Chairs · 2023-10-07

**Decision:**

Accept-Main

**Comment:**

This paper proposes the IBKD method in order to make pre-trained language models (PLMs) more accessible by distilling them into smaller representation models. The topic studied is an interesting topic and the proposed method based on information bottleneck is meaningful. Reviewers have concerns on the insufficient experiments or missing details. During the rebuttal period, the authors have replied to these concerns with more details and experimental results, which lead to overall positive scores from reviewers.